# Determination of the Diagnostic Performance of Laboratory Tests in the Absence of a Perfect Reference Standard: The Case of SARS-CoV-2 Tests

**DOI:** 10.3390/diagnostics13182892

**Published:** 2023-09-09

**Authors:** Sonja Hartnack, Henning Nilius, Sabrina Jegerlehner, Franziska Suter-Riniker, Pascal Bittel, Philipp Jent, Michael Nagler

**Affiliations:** 1Section of Epidemiology, Vetsuisse Faculty, University of Zurich, 8057 Zuric, Switzerland; 2Department of Clinical Chemistry, Inselspital, Bern University Hospital, University of Bern, 3010 Bern, Switzerland; henning.nilius@insel.ch (H.N.); michael.nagler@insel.ch (M.N.); 3Department of Emergency Medicine, Inselspital, Bern University Hospital, 3010 Bern, Switzerland; sabrina.jegerlehner@insel.ch; 4Institute for Infectious Diseases, University of Bern, 3010 Bern, Switzerland; franziska.suter@unibe.ch (F.S.-R.); pascal.bittel@unibe.ch (P.B.); 5Department of Infectious Diseases, Inselspital, Bern University Hospital, University of Bern, 3010 Bern, Switzerland; philipp.jent@insel.ch

**Keywords:** BLCM, diagnostic accuracy, no gold standard models

## Abstract

Background: Currently, assessing the diagnostic performance of new laboratory tests assumes a perfect reference standard, which is rarely the case. Wrong classifications of the true disease status will inevitably lead to biased estimates of sensitivity and specificity. Objectives: Using Bayesian’ latent class models (BLCMs), an approach that does not assume a perfect reference standard, we re-analyzed data of a large prospective observational study assessing the diagnostic accuracy of an antigen test for the diagnosis of SARS-CoV-2 infection in clinical practice. Methods: A cohort of consecutive patients presenting to a COVID-19 testing facility affiliated with a Swiss University Hospital were recruited (n = 1465). Two real-time PCR tests were conducted in parallel with the Roche/SD Biosensor rapid antigen test on nasopharyngeal swabs. A two-test (PCR and antigen test), three-population BLCM was fitted to the frequencies of paired test results. Results: Based on the BLCM, the sensitivities of the RT-PCR and the Roche/SD Biosensor rapid antigen test were 98.5% [95% CRI 94.8;100] and 82.7% [95% CRI 66.8;100]. The specificities were 97.7% [96.1;99.7] and 99.9% [95% CRI 99.6;100]. Conclusions: Applying the BLCM, the diagnostic accuracy of RT-PCR was high but not perfect. In contrast to previous results, the sensitivity of the antigen test was higher. Our results suggest that BLCMs are valuable tools for investigating the diagnostic performance of laboratory tests in the absence of perfect reference standard.

## 1. Introduction

During the COVID-19 pandemic, numerous diagnostic tests for the detection of SARS-CoV-2 in patient samples have been developed. RT-PCR assays were among the first diagnostic tests to be developed [1,2]. Soon, it became evident that false-negative results could occur [3]. Rapid antigen tests were developed to improve the limited laboratory capacities, accessibility, and affordability. The diagnostic accuracy of these newly developed rapid antigen tests has been evaluated numerous times by considering RT-PCR as the gold standard. If the RT-PCR is not 100% sensitive (not classifying all truly infected patients as such) or not 100% specific (not classifying all truly negative patients as such), then evaluating a rapid antigen test by comparing its test results with RT-PCR results inevitably leads to a biased evaluation of the new test [4]. Different RT-PCR sensitivities were also found for in- and outpatients [5]. The classical understanding of diagnostic sensitivities and specificities as “intrinsic” test characteristics has become obsolete as sensitivities and specificities are assumed to vary with external factors [6].

Since the pioneering work from [7]—after whom is named the Hui-Walter paradigm—on estimating diagnostic test accuracies in the absence of a (perfect) gold standard, further developments took place, including Bayesian latent class models (BLCMs) [8] and user-friendly software for MCMC (Markov Chain Monte Carlo) simulation [9]. Specific STARD reporting guidelines for BLCMs have been developed [10]. Although, from a public health perspective, it would be highly relevant to have robust estimates of diagnostic test accuracies in a given population, very few attempts to use BLCMs to assess the diagnostic performance of COVID-19 tests exist, i.e., [11,12,13,14].

This study aimed to re-analyze data from a prospective cohort study evaluating a rapid antigen test in a real-life clinical setting, thus exploring the utility of BLCMs in such a setting [15].

## 2. Materials and Methods

### 2.1. Study Design, Patients, and Population

Using a Bayesian latent class approach, we re-analyzed the data [15] of a recent prospective cohort study. This dataset consists of dichotomized test results from two diagnostic tests of 1462 consecutive patients enrolled in a prospective cross-sectional study conducted from January to March 2021. The following inclusion criteria were applied: (1) suspicion of SARS-CoV-2 infection, (2) age 18 years or older, and (3) signed informed consent. The flow of the patients is given in Figure 1. The COVID-19 testing facility, affiliated with Inselspital University Hospital, and a specialized hospital employing high-throughput RT-PCR [16], was one of Switzerland’s largest facilities [17]. Following the instructions of the authorities, patients appeared for three different reasons: (a) symptoms consistent with SARS-CoV-2 infection, (b) contact with infected patients, and (c) other reasons such as travel certificates or intended shortening of quarantine. From the 1465 patients presenting to the testing facility three patients had to be excluded from this analysis as they had insufficient material to perform the antigen test. The local ethics committee reviewed and approved the protocol (Bern Cantonal Ethics Committee #2020-02729). All individuals have signed an informed consent form, and the study was conducted in accordance with the Declaration of Helsinki.

### 2.2. Study Processes and Determination of Laboratory Test

Before the consultation, the individuals were informed by trained medical staff. Patients completed a questionnaire that was prepared in accordance with the official requirements in Switzerland. A specially trained physician checked the questionnaire’s answers and followed up in case of doubt. A specially trained nurse performed the nasopharyngeal swabs according to a protocol that follows official guidelines. Liofilchem Viral Transport Medium (Roseto degli Abruzzi, Italy) and iClean Specimen Collection Flocked Swabs (Cleanmo Technology Co., Shenzhen, China) were used. The sample material was at 4 °C and performed within 6 h (antigen test) or 12 h (RT-PCR). The coded clinical data and laboratory test results were kept in different databases and merged only after analysis. All details have been described in detail previously [15].

Both the Roche/SD Biosensor SARS-CoV-2 antigen test (Roche Diagnostics, Mannheim, Germany) and RT-PCR were performed from the same sample material by a trained medical laboratory technician. The manufacturer’s specifications were strictly followed, and internal quality controls were performed daily (package leaflet). Details have been given elsewhere [15]. Two real-time PCRs were performed (Roche cobas^®^ SARS-CoV-2; Seegene Allplex 2019-nCoV), following the manufacturers’ instructions, on a STARlet IVD System or a cobas 8800 system, as has been previously described [16,18]. RT-PCR was performed as part of the daily routine without the laboratory technicians knowing the antigen test result. Commercial internal quality controls were carried out with each run. A cycle threshold of 40 was considered positive. The frequencies of the dichotomized test results of the antigen test and the PCR are displayed in Table 1.

### 2.3. Bayesian Latent Class Modeling

In the absence of a gold standard test, a Bayesian latent class model (BLCM) was fitted to the data to obtain estimates of sensitivity and specificity for each of the tests and prevalence in the population.

For the BLCM, we followed the classical Hui Walter approach [7] with two diagnostic tests and three populations (symptoms, exposure, and other). With dichotomized test results from two tests in three populations, the independence model without any conditional dependency is fully identifiable because there are nine degrees of freedom (i.e., three from each population) and seven parameters to be estimated: sensitivities and specificities of both tests and prevalences of each population. The four possible combinations dichotomized test results (−−,+−, −+, ++) in the three populations (*i*) are assumed to follow an independent multinomial distribution:$$y_{i}\sim Multinomial(n_{i},\left(p_{i_{--}},p_{i_{+-}},p_{i_{-+}},p_{i_{++}}\right))$$
with the following four multinomial cell probabilities:$$p_{i_{--}}={pr}_{i}*\left(\left(1-{Se}_{AT}\right)*\left({1-Se}_{PCR}\right)+{cse}_{12}\right)+\left(1-{pr}_{i}\right)*(\left({Sp}_{AT}\right)*\left({Sp}_{PCR}\right)+{csp}_{12})\;p_{i_{+-}}={pr}_{i}*\left(\left({Se}_{AT}\right)*\left(1-{Se}_{PCR}\right)-{cse}_{12}\right)+\left(1-{pr}_{i}\right)*(\left(1-{Sp}_{AT}\right)*\left({Sp}_{PCR}\right)-{csp}_{12})\;p_{i_{-+}}={pr}_{i}*\left(\left(1-{Se}_{AT}\right)*\left({Se}_{PCR}\right)-{cse}_{12}\right)+\left(1-{pr}_{i}\right)*(\left({Sp}_{AT}\right)*\left({1-Sp}_{PCR}\right)-{csp}_{12})\;p_{i_{++}}={pr}_{i}*\left(\left({Se}_{AT}\right)*\left({Se}_{PCR}\right)+{cse}_{12}\right)+\left(1-{pr}_{i}\right)*(\left({1-Sp}_{AT}\right)*\left(1-{Sp}_{PCR}\right)+{csp}_{12})$$

Here, *pr* is the prevalence and *i* indicates one of the three populations. *Se* and *Sp* are the sensitivities and specificities of the two tests, and *cse*_12_ and *csp*_12_ are the conditional dependencies between either the two sensitivities or the two specificities.

We used a Bayesian estimation framework with beta distributions Be(a,b) for the parameters of interest, i.e., sensitivities, specificities, and prevalences. We chose an informative prior for the specificity of the PCR, assuming that we are 95% sure that this specificity is higher than 90%, with a mode at 99%. Using betabuster, this assumption leads to dbeta(34.166327738, 1.335013089). For the other parameters of interest, we chose minimally informative priors dbeta(1,1). To assess the potential effect of these priors on the posterior estimates, we performed a sensitivity analysis with different combinations of alternative priors, mainly weakly informative priors with dbeta(2,1), but also informative priors designating a high sensitivity of PCR and a low prevalence in those which had no symptoms or contact with an infected person (population other) (Table 2).

We used Markov Chain Monte Carlo simulations with Gibbs sampling with JAGS version 4.3.0 (Plummer, 2003). The package runjags [19] was used to access JAGS and to prepare the model code. The model and the data are available in the Appendix A (S1.model.code.data.bug).

## 3. Results

The latent class model converged based on the visual inspection of the trace plots and the Gelman–Rubin statistic. The estimates for the sensitivities and specificities of both tests and the three prevalences are given in Figure 2 and Table 3. Both the sensitivity and specificity of the RT-PCR assay are close to 100%. There was no evidence for a conditional dependency between sensitivities nor specificities, as evidenced by 95% credibility intervals, which included 0. Whilst the specificity of the rapid antigen test is close to 100%, the sensitivity is considerably lower and—compared to the other parameters of interest—the 95% credibility intervals are wider.

When performing a sensitivity analysis by changing the prior information, the only posterior estimate that changed substantially with nearly 3% (absolute values) was the sensitivity of the rapid antigen test. As expected, the prevalence in patients with symptoms or exposure to COVID-19 cases was higher than in patients who came for other testing reasons.

## 4. Discussion

We used data from a large prospective cohort study analyzing the accuracy of a rapid antigen SARS-CoV-2 antigen test in clinical practice to apply a BLCM [15]. The resulting sensitivity of the rapid antigen test was considerably higher (82.7; 95% CRI 66.8; 100%) compared to the original analysis (65.3; 95% CRI 56.8; 73.1%). This finding might be explained by the diagnostic accuracies of the reference standard RT-PCR being close, but not equal, to 100%. In contrast, the specificity of the rapid antigen test is similarly high in the gold standard and the BLCM analysis.

To our knowledge, this is the first study applying a BLCM to estimate the diagnostic accuracy of the Roche/SD Biosensor rapid antigen test. The strength of our analysis is that we have included a large number of consecutive patients in clinical practice. The prospective design employing a strict protocol ensured complete and accurate data. Limitations of this study include the unequal sample sizes of the three populations, so the largest population, patients with symptoms, was possibly influenced the posteriors more than the two other populations. Additionally, although the model is theoretically identifiable with two diagnostic tests and three populations, the difference in the prevalence of the three populations is less than 10%, which might affect the estimates’ precision [20]. Furthermore, the assumption of constant diagnostic sensitivity in the three populations is also questionable. This is possibly the reason for the wider credibility intervals of the rapid antigen compared to the credibility intervals of the other parameters of interest. Possibly, in the considered sample of patients, different subpopulations exist, e.g., depending on the presence of clinical symptoms, more or fewer virus copies might be present, which might have a more considerable impact on the sensitivity of the rapid antigen test compared to RT-PCR.

The results of this study suggest that BLCMs are valuable tools when there is an imperfect reference standard, including for diagnosing respiratory infections in a routine clinical setting. The higher sensitivity of the antigen test could well be explained by the fact that the reference standard (RT-PCR) is not entirely accurate. Unfortunately, the “true value” cannot be determined empirically. Each method has its assumptions, which are not strictly adhered to, and which requirements are more significant in the situation at hand is difficult to say. BLCMs, similar to the classical gold standard approach, rely on the assumption that a positive and a negative test result indicates the same infection status for both tests.

This can be done already in the project’s protocol phase, thus supporting data analysis. Either separate BLCMs are conducted for patients with and without symptoms, or the symptoms (with/without) are included as a covariate in the evaluation of the diagnostic test [21]. We agree with [22] that BLCMs are complex models, that conditional dependencies are important, and that BLCMs are challenging to interpret by readers without a statistical background. We also agree with the famous statement of George Box: “all models are wrong, but some are useful” [23]. Still, in a pandemic situation, when new diagnostic tests are urgently needed, inter- and transdisciplinary research should be conducted, and a lack of statistical knowledge should not preclude the application of useful models.

## 5. Conclusions

In conclusion, BLCMs provide a valuable approach for determining diagnostic accuracy measures in the presence of an imperfect reference standard. This applies also to diagnosing respiratory infections in a routine clinical setting. Key limitations of this method can already be addressed in the planning phase of diagnostic accuracy studies.

## Figures and Tables

**Figure 1 diagnostics-13-02892-f001:**
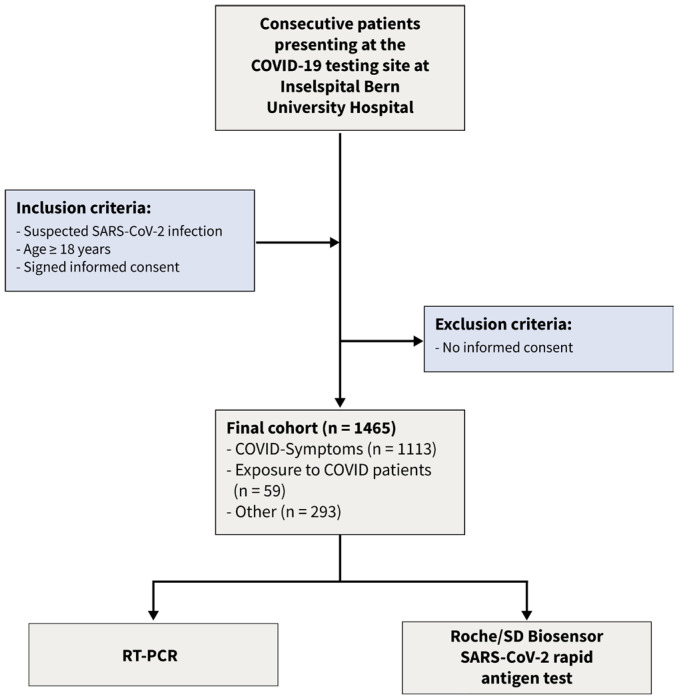
Flow of the patients.

**Figure 2 diagnostics-13-02892-f002:**
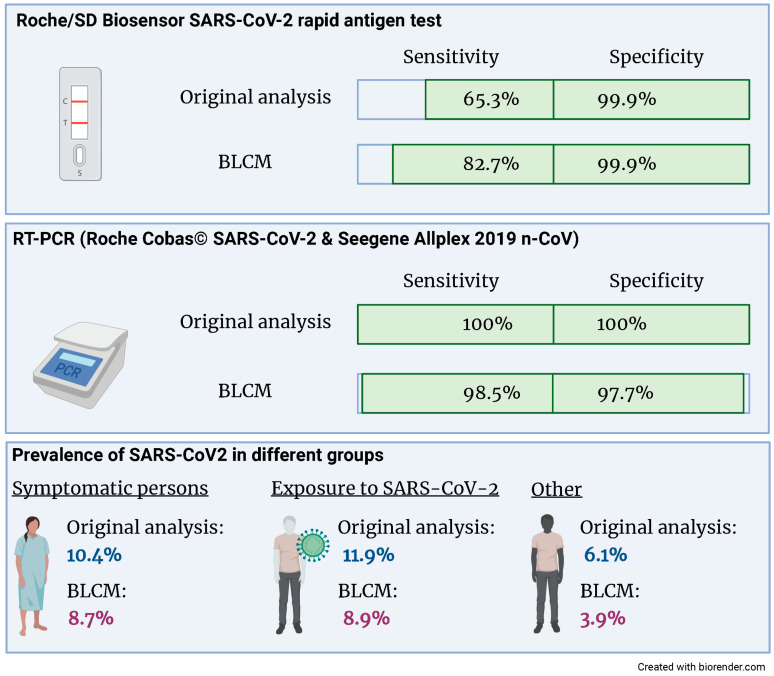
Compared to the original analysis, the sensitivity of the Roche/SD Biosensor SARS-CoV-2 rapid antigen test increased from 65.3% to 82.7% when accounting for an imperfect gold standard. Also, the prevalence of SARS-CoV2 was reduced in all three groups.

**Table 1 diagnostics-13-02892-t001:** Frequencies of the respective dichotomized test results (Roche/SD Biosensor antigen test, RT-PCR) cross-classified in patients with symptoms with exposition to COVID-19 cases and patients being tested for other reasons. − negative test results; + positive test results.

Tests		Populations		
Antigen Test	RT-PCR	Symptoms	Exposition	Other
−	−	993	52	247
+	−	1	0	1
−	+	35	4	10
+	+	81	3	8
		Total: 1110	Total: 59	Total: 293

**Table 2 diagnostics-13-02892-t002:** Sensitivity analysis with different combinations of alternative priors.

	a	b	c	d	e	f	g
Antigen Test							
Sensitivity	84.4 [67.1;1]	82.8 [66.6;1]	84.2 [67.7;1]	85.3 [68.0;1]	82.7 [66.8;1]	83.7 [67.7;1]	83.4 [67.1;1]
Specificity	99.9 [99.6;1]	99.9 [99.6;1]	99.9 [99.6;1]	99.9 [99.6;1]	99.9 [99.6;1]	99.9 [99.6;1]	99.8 [99.6;1]
**RT-PCR**							
Sensitivity	98.5 [94.9;1]	98.5 [95.0;1]	98.5 [94.8;1]	98.7 [95.9;1]	98.6 [95.9;1]	98.7 [95.9;1	98.0 [92.7;1]
Specificity	97.6 [96.0;99.8]	97.7 [96.1;99.7]	97.6 [96.0;99.6]	97.5 [95.9;99.6]	97.8 [96.1;99.7]	97.7 [96.0;99.6]	97.6 [96.0;99.5]
**Prevalence 1**							
	8.6 [6.5;11.1]	8.7 [6.6;11.1]	8.7 [6.5;11.1]	8.5 [6.4;10.9]	8.8 [6.6;11.2]	8.7 [6.6;11.0]	8.7 [6.6;11.2]
**Prevalence 2**							
	8.7 [1.9;17.7]	8.8 [1.8;17.9]	8.7 [1.9;17.6]	8.4 [1.6;6.3]	8.9 [1.8;17.9]	8.7 [1.8;17.7]	8.7 [1.9;17.7]
**Prevalence 3**							
	3.9 [1.5;7.1]	4.0 [1.6;7.2]	3.9 [1.5;7.1]	3.7 [1.6;6.3]	4.0 [1.6;7.2]	3.8 [1.7;6.5]	3.7 [1.6;6.4]

^a^ Minimally informative priors for all parameter (dbeta(1,1)). ^b^ Informative prior for specificity PCR (“95% sure that sp PCR is greater than 90% with a mode at 99%” with a beta(a = 34.166327738, b = 1.335013089), weakly informative prior for se PCR (dbeta(2,1), minimally informative prior for all other parameters (dbeta(1,1)). ^c^ Informative prior for specificity PCR (“95% sure that sp PCR is greater than 90% with a mode at 99%” with a beta(a = 34.166327738, b = 1.335013089), weakly informative prior for se and sp Antigen test (dbeta(2,1), minimally informative prior for all other parameters (dbeta(1,1)). ^d^ Informative prior for sensitivity PCR (“95% sure that se PCR is greater than 90% with a mode at 99%” with a beta(a = 34.166327738, b = 1.335013089), minimally informative prior for all other parameters (dbeta(1,1)). ^e^ Informative prior for se and sp PCR (“95% sure that se PCR and sp PCR are greater than 90% with a mode at 99%” with a beta(a = 34.166327738, b = 1.335013089, minimally informative prior for all other parameters (dbeta(1,1)). ^f^ Informative priors for se and sp PCR (“95% sure that se PCR and sp PCR are greater than 90% with a mode at 99%” with a beta(a = 34.166327738, b = 1.335013089), informative prior for prevalence in patients with other reasons than symptoms or exposition (I am 95% sure that the prevalence in the group “other” is below 10% with a mode at 3% (dbeta(2.6262,53.5809)), minimally informative prior for all other parameters (dbeta(1,1)). ^g^ Covariance term between the sensitivities of both tests included. Informative priors for se and sp PCR (“95% sure that se PCR and sp PCR are greater than 90% with a mode at 99%” with a beta(a = 34.166327738, b = 1.335013089), informative prior for prevalence in patients with other reasons than symptoms or exposition (I am 95% sure that the prevalence in the group “other” is below 10% with a mode at 3% (dbeta(2.6262,53.5809)), minimally informative prior for all other parameters (dbeta(1,1)).

**Table 3 diagnostics-13-02892-t003:** Posterior medians and 95% credibility intervals for the diagnostic sensitivities (Se) and specificities (Sp) for the Roche/SD Biosensor rapid antigen test and the RT-PCR and the three prevalences for patients with symptoms, patients being exposed to COVID-19 cases, and patients being tested for other reasons.

	Median	95% CI
Sensitivity antigen test	82.7	[66.8;100]
Specificity antigen test	99.9	[99.6;100]
Sensitivity RT-PCR	98.5	[94.8;100]
Specificity RT-PCR	97.7	[96.1;99.7]
Prevalence 1 (symptoms)	8.7	[6.6;11.2]
Prevalence 2 (exposition)	8.9	[1.8;17.7]
Prevalence 3 (other)	3.9	[1.6;7.2]

## Data Availability

The dataset and the code are available in the Appendix A (S.model.code.data.bug).

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
