# Peer review of "Determination of the Diagnostic Performance of Laboratory Tests in the Absence of a Perfect Reference Standard: The Case of SARS-CoV-2 Tests"

_diagnostics, 2023, doi:10.3390/diagnostics13182892_

Round 1

Reviewer 1 Report

The authors using Bayesian’ latent class models (BLCM) for assessment the sensitivity and specific of two test (real-time PCR and  Roche/SD Biosensor rapid antigen test) in the diagnosis of SARS-CoV-2 infection, where the RT-PCR was considered the gold standard. However, RT-PCR is not 100% sensitive (not classifying all truly infected patients as such), and / or not 100% specific (not classifying all truly negative patients as such).

The aim of this study was to re-analyse data from a prospective cohort study evaluating a rapid antigen test in a real-life clinical setting, thus exploring the utility of BLMC 62 in such a setting.

The results of this study suggest that BLMC models are useful tools when there is an imperfect reference standard, including for diagnosing respiratory infections in a routine clinical setting.

The use of BLMC models will allow know the gold standard diagnostic method that serves as reference to compare new diagnostic tests.

Author Response

We thank the reviewer for the positive appraisal. We agree that the applications of BLCMs to assess the performance of diagnostic tests is particularly useful in routine clinical settings.

--------------------------------------------------------

Reviewer 2 Report

This study re-analysed data using Bayesian’ latent class models from a prospective cohort study evaluating a rapid antigen test in a real-life clinical setting. The writing is clear and the results provide further evidence on the accuracy of the rapid antigen test. However, there is no much new information because a mature analysis method is used. The authors should also add some description on what is the reference method for determining the sensitivity and the specificity of the RT-PCR methods and the rapid antigen test.

English is good except that some sentences need adjusting for better clarity. For example, Line 236-237, Either separate BLCM are run for patients with and without symptoms, or having symptoms is included as a covariate in the diagnostic test evaluation (Lewis and Torgerson, 2012). 

Author Response

We thank the reviewer for the positive appraisal. We agree that the analysis method is well established and even endorsed by WOAH (World Organization of Animal Health). However, although it is a mature analytical method, it is extremely rarely used in the evaluation of diagnostic tests in humans and especially not in the case of COVID-19. So, the novelty of our approach lies in the application of BLCMs for Covid-19 tests.

Point 1: The authors should also add some description on what is the reference method for determining the sensitivity and the specificity of the RT-PCR methods and the rapid antigen test.

RESPONSE 1: We are not sure if we understand the comment of the reviewer. The idea of the BLCM modelling to circumvent the lack of a reference method which is 100% sensitive and specific. The BLCM approach is also called “no gold standard models”.

Point 2: English is good except that some sentences need adjusting for better clarity. For example, Line 236-237, Either separate BLCM are run for patients with and without symptoms, or having symptoms is included as a covariate in the diagnostic test evaluation (Lewis and Torgerson, 2012).

RESPONSE 2: Thank you for spotting. We checked the whole manuscript for sentences that were difficult to understand and made some improvements.

--------------------------------------------------------

Reviewer 3 Report

1; This study does not provide significant information.

2: The provided data can be expanded and presented perfectly.

3; The authors have shown positive by RT-PCR but negative by Antigen test and vice versa. Generally, RT-PCR gives better result.

4: This study has very small sample and should be expanded to come better conclusion.

Minor english editing is required.

Author Response

Point 1: This study does not provide significant information.

RESPONSE 1: We thank the reviewer for the possibility to clarify this point. With the following manuscript, we aimed to address the problem of imperfect reference standards (page 1 line 20 to line 22; page 2 line 61 to line 63). The existence of perfect reference tests (100% sensitivity and 100% specificity) is a basic assumption of today's scientific methodology for the evaluation of diagnostic tests (doi: 10.1016/j.jclinepi.2009.02.005). However, this does not correspond to reality and even small deviations lead to biased estimates of diagnostic quality. Therefore, new methods for determining test performance are needed. In the present manuscript, we applied a new analytical method – the Bayesian latent class approach (BLCM; page 2, line 53 to line 60) - to the situation of diagnosis of SARS-CoV-2 infection. This setting is particularly suitable because (a) several studies have indicated that RT-PCR for SARS-CoV-2 has a number of limitations and (b) methodologically well-done studies are available (page 2 line 39 to line 51). We used data of a recent, very carefully planned and executed prospective cohort study including consecutive individuals presenting at a COVID-19 testing facility (page 2, line 64 to 79; https://doi.org/10.1016/j.ijid.2021.07.010). With 1,465 patients, this was one of the largest clinical studies to adequately reflect conditions in clinical practice. The results have generated substantial attention in terms of citations and media coverage (https://www.ijidonline.com/article/S1201-9712(21)00565-8/fulltext).

In the current manuscript, we describe the BLCM methodology in detail (page 4 line 108 to line 147), present the results in several tables and texts (line 175 to line 199), and discuss the results and the implications in the discussion part (line 200 to line 243). In particular, a graphical abstract (Figure 2) illustrates the results to the results of the original analysis. Therefore, we are convinced that our manuscript contains sufficient and significant information on the background, methodology, results, limitations and implications. We hope that the reviewer will follow us here.

Point 2: The provided data can be expanded and presented perfectly.

RESPONSE 2: Thank you for raising this issue. Within the present manuscript – submitted as a “brief report” - we provide detailed information regarding the study design, patients and population (line 65 to line 79), flow of the patients (Figure 1), study processes and determination of laboratory tests (line 81 to line 100), reference to the previous publication stating more methodological details, frequencies of the test results (Table 1), overall results (Table 3; line 175 to line 184), a graphical illustration (Figure 2), sensitivity analyses (Table 2, line 191 to line 195), and the data set and code (supplementary material).

Point 3: The authors have shown positive by RT-PCR but negative by Antigen test and vice versa. Generally, RT-PCR gives better result.

RESPONSE 3: Thank you for raising this important issue. We discussed the results of the original analysis and the results of the BLCM analysis in the discussion part of the manuscript (line 200 to line 243).

Point 4: This study has very small sample and should be expanded to come better conclusion.

RESPONSE 4: Thank you for the possibility to clarify this point. Various studies analyzing the diagnostic accuracy of rapid antigen tests have been conducted, with a systematic review conducted by the Cochrane Collaboration summarizing these data (Dinnes et al., 2020 ). The authors raised major methodological concerns and a considerable risk of bias in all previous studies. In particular, the applicability was estimated to be low because of biased patient selection. In contrast to these studies, ours paid close attention to all the requirements of diagnostic accuracy studies: (a) an adequately powered prospective cross-sectional design examining a clearly defined clinical question; (b) selection of an appropriate study population (real-life clinical setting); (c) accurate determination of the index test; (d) rigorous choice and determination of the reference standard test; and (e) optimal flow and timing. We believe that this difference in study design and methodological quality explains the significant differences in sensitivities obtained. This distinguishes our study from many others. Compared to very many other studies, a number of 1,465 included patients is relatively high. We hope that the reviewer can follow us here. 

Point 5: Minor english editing is required.

RESPONSE 5: Thank you for spotting. We checked the whole manuscript for sentences that were difficult to understand and made some improvements.

=======================================

Round 2

Reviewer 3 Report

Introduction should be improved

materials and methods should be expanded.

discussion and conclusion can be expanded a little more.

limitation of the study should be discussed

Author should revised the MS as suggested

Author Response

Point

  1. Introduction should be improved
  2. materials and methods should be expanded.
  3. discussion and conclusion can be expanded a little more.
  4. limitation of the study should be discussed

RESPONSE: We thank the reviewer and the editor for their valuable time devoted to the intensive review of our manuscript, which we submitted as a "brief report." These reviews are precious to improve the quality of the manuscript.

We must confess that it is difficult to understand the requested improvements. The comments are very general, and we have extensively responded to the already generic first-round comments (given below). In the following, we will summarize the contents of the sections mentioned by the reviewer and kindly ask for specific guidance. (1) In the introductory chapter, we outlined the problem posed by imperfect reference standards for traditional analysis of diagnostic accuracy studies. We further delineated why this presented a problem, especially concerning diagnostic tests for COVID-19. Moreover, we discussed why Bayesian latent class models (BLCM) might be a solution to the problem. Finally, we formulated the aim of the study, applying the analytical technique to an existing, high-quality dataset of a prospective cohort study. (2) In the methods section, we detailed the study design, patients and population covering the setting, the inclusion criteria, population characteristics, and details of ethical approval. Figure 1 gives a PRISMA flow sheet. We provided all details regarding study processes and the determination of laboratory tests. We reference the original publication, discussing the prospective cohort study in more detail. Furthermore, we give extensive details and references concerning the BLCM analytical technique. (3, 4) We provide a complete discussion section covering (a) a paragraph summarizing our findings in words, (b) discussing our results in the context of previous literature, (c) discussing several limitations of the study, (d) give an interpretation in terms of "what does it mean for scientific inquiry," and (e) provide a conclusion.

First round comments

Comment 1: This study does not provide significant information.

RESPONSE: We thank the reviewer for the possibility to clarify this point. With the following manuscript, we aimed to address the problem of imperfect reference standards (page 1 line 20 to line 22; page 2 line 61 to line 63). The existence of perfect reference tests (100% sensitivity and 100% specificity) is a basic assumption of today's scientific methodology for the evaluation of diagnostic tests (doi: 10.1016/j.jclinepi.2009.02.005). However, this does not correspond to reality and even small deviations lead to biased estimates of diagnostic quality. Therefore, new methods for determining test performance are needed. In the present manuscript, we applied a new analytical method – the Bayesian latent class approach (BLCM; page 2, line 53 to line 60) - to the situation of diagnosis of SARS-CoV-2 infection. This setting is particularly suitable because (a) several studies have indicated that RT-PCR for SARS-CoV-2 has a number of limitations and (b) methodologically well-done studies are available (page 2 line 39 to line 51). We used data of a recent, very carefully planned and executed prospective cohort study including consecutive individuals presenting at a COVID-19 testing facility (page 2, line 64 to 79; https://doi.org/10.1016/j.ijid.2021.07.010). With 1,465 patients, this was one of the largest clinical studies to adequately reflect conditions in clinical practice. The results have generated substantial attention in terms of citations and media coverage (https://www.ijidonline.com/article/S1201-9712(21)00565-8/fulltext).

In the current manuscript, we describe the BLCM methodology in detail (page 4 line 108 to line 147), present the results in several tables and texts (line 175 to line 199), and discuss the results and the implications in the discussion part (line 200 to line 243). In particular, a graphical abstract (Figure 2) illustrates the results to the results of the original analysis. Therefore, we are convinced that our manuscript contains sufficient and significant information on the background, methodology, results, limitations and implications. We hope that the reviewer will follow us here.

Comment 2: The provided data can be expanded and presented perfectly.

RESPONSE: Thank you for raising this issue. Within the present manuscript – submitted as a “brief report” - we provide detailed information regarding the study design, patients and population (line 65 to line 79), flow of the patients (Figure 1), study processes and determination of laboratory tests (line 81 to line 100), reference to the previous publication stating more methodological details, frequencies of the test results (Table 1), overall results (Table 3; line 175 to line 184), a graphical illustration (Figure 2), sensitivity analyses (Table 2, line 191 to line 195), and the data set and code (supplementary material).

Comment 3: The authors have shown positive by RT-PCR but negative by Antigen test and vice versa. Generally, RT-PCR gives better result.

RESPONSE: Thank you for raising this important issue. We discussed the results of the original analysis and the results of the BLCM analysis in the discussion part of the manuscript (line 200 to line 243).

Comment 4: This study has very small sample and should be expanded to come better conclusion.

RESPONSE: Thank you for the possibility to clarify this point. Various studies analyzing the diagnostic accuracy of rapid antigen tests have been conducted, with a systematic review conducted by the Cochrane Collaboration summarizing these data (Dinnes et al., 2020 ). The authors raised major methodological concerns and a considerable risk of bias in all previous studies. In particular, the applicability was estimated to be low because of biased patient selection. In contrast to these studies, ours paid close attention to all the requirements of diagnostic accuracy studies: (a) an adequately powered prospective cross-sectional design examining a clearly defined clinical question; (b) selection of an appropriate study population (real-life clinical setting); (c) accurate de- termination of the index test; (d) rigorous choice and determination of the reference standard test; and (e) optimal flow and timing. We believe that this difference in study design and methodological quality explains the significant differences in sensitivities obtained. This distinguishes our study from many others. Compared to very many other studies, a number of 1,465 included patients is relatively high. We hope that the reviewer can follow us here. 

Comment 5: Minor english editing is required.

RESPONSE: Thank you for spotting. We checked the whole manuscript for sentences that were difficult to understand and made some improvements.